# Development of Immersion and Oral Bivalent Nanovaccines for Streptococcosis and Columnaris Disease Prevention in Fry and Fingerling Asian Seabass (*Lates calcarifer*) Nursery Farms

**DOI:** 10.3390/vaccines12010017

**Published:** 2023-12-22

**Authors:** Pakapon Meachasompop, Anurak Bunnoy, Wisawat Keaswejjareansuk, Piroonrat Dechbumroong, Katawut Namdee, Prapansak Srisapoome

**Affiliations:** 1Laboratory of Aquatic Animal Health Management, Department of Aquaculture, Faculty of Fisheries, Kasetsart University, 50 Paholayothin Rd., Ladyao, Chatuchak, Bangkok 10900, Thailand; pakapon.mea@ku.th (P.M.); ffisarb@ku.ac.th (A.B.); 2Center of Excellence in Aquatic Animal Health Management, Faculty of Fisheries, Kasetsart University, 50 Paholayothin Rd., Ladyao, Chatuchak, Bangkok 10900, Thailand; 3National Nanotechnology Center (NANOTEC), National Science and Technology Development Agency (NSTDA), Thailand Science Park, Pathumthani 12120, Thailand; keaswejjareansuk@wpi.edu (W.K.); piroonrat.d@gmail.com (P.D.); katawut@nanotec.or.th (K.N.)

**Keywords:** Asian seabass, streptococcosis, columnaris, bivalent nanovaccines, immune responses, disease resistance

## Abstract

In the present study, chitosan-based bivalent nanovaccines of *S. iniae* and *F. covae* were administered by immersion vaccination at 30 and 40 days after hatching (DAH), and the third vaccination was orally administered by feeding at 50 DAH. ELISA revealed that the levels of total IgM and specific IgM to *S. iniae* and *F. covae* were significantly elevated in all vaccinated groups at 10, 20, and 30 days after vaccination (DAV). A qRT-PCR analysis of immune-related genes revealed significantly higher *IgT* expression in the vaccinated groups compared to the control group, as revealed by 44–100-fold changes in the vaccinated groups compared to the control (*p* < 0.001) at every tested time point after vaccination. All vaccinated groups expressed *IgM*, *MHCIIα*, and *TCRα* at significantly higher levels than the control group at 10 and/or 20 DAV (*p* < 0.05). In the *S. iniae* challenge tests, the survival of vaccinated groups ranged from 62.15 ± 2.11 to 75.70 ± 3.36%, which significantly differed from that of the control group (44.44 ± 1.92%). Similarly, all vaccinated groups showed higher survival rates of 68.89 ± 3.85 to 77.78 ± 5.09% during *F. covae* challenge than the control groups (50.00 ± 3.33%) (*p* < 0.05).

## 1. Introduction

Asian seabass (*Lates calcarifer*) is a euryhaline fish indigenous to the Indo-Pacific region and is extensively farmed in southeast Asia, notably in Thailand. Its capacity to survive in fresh and saltwater environments makes it an essential aquacultural commodity in Thailand, with significant economic value. The growth of the Asian seabass farming industry in the region has been propelled by this high demand, leading to a substantial expansion of its cultivation. The cultivation of Asian seabass has not only enhanced its domestic popularity in Thailand but has also resulted in exportation to countries such as China, Taiwan, Vietnam, Singapore, and Malaysia [1,2,3]. However, the Asian seabass farming industry is facing problems due to the outbreak of diseases such as bacterial and viral infections, which are causing significant damage to fish farms. Among those harmful diseases, streptococcosis and columnaris infections, which are caused by *Streptococcus iniae* and *Flavobacterium covae*, respectively, have been identified as major causative agents that significantly hamper Asian seabass aquaculture [3,4,5,6].

To overcome disease problems in fish aquaculture, vaccines are considered one of the most effective methods for disease prevention, acting as an alternative to chemical and drug treatment methods, which always have many adverse side effects both in the environment and for consumers [7]. Unlike in higher vertebrates, vaccination methods in fish can be conducted through injection, immersion, and oral administration. Of these, injection routes are classified as the most effective for immune responses and protection. However, these methods are time- and cost-consuming and always induce stress and mortality after vaccination [8]. Additionally, this method is too difficult to booster vaccinate in the nursery or juvenile to adults during culture in the grow-out stages. Therefore, immersion and oral administration methods are intensively considered because these techniques are easy to perform in small fish, and a high number of fish can be simultaneously vaccinated in hatcheries or nurseries, and booster vaccination can also be more practical [9,10].

Compared to the injection route, immersion and oral vaccination have relatively low immune responses and protection efficacy [11]. To increase and meet the better efficacy of oral and immersion vaccines, nanotechnology is a novel technique developed to enhance vaccine efficacy in various fish species [11,12,13,14]. This technology involves reducing the size of vaccine particles, enabling them to adhere more efficiently to various mucosa-associated lymphoid tissues (MALTs), which effectively exert immune functions in the skin, nasal, gill, and gastrointestinal tract of fish, thus promoting better and longer-lasting immune responses and protection via locally specific immunoglobulin (Ig) production [14,15]. However, since various diseases occur at different times throughout the production cycle, administering bi- or polyvalent vaccines over the culture period can be targeted to effectively increase the quantity and quality of fish production and reduce production costs, which are preferable for fish farmers. Therefore, bivalent or multivalent vaccines would be a more attractive alternative to effectively enhance the specific immune system of fish at early developmental stages, where the immune system is already immunocompetent [12,15].

Based on the current information, various bivalent nanovaccines have been developed and have shown excellent immune responses and protection in different fish species, including bivalent vaccines against columnaris and francisellosis in Nile tilapia [12].

At present, however, there are no reports on using nanovaccines in the aquaculture industry to prevent diseases, especially streptococcosis and columnaris, which are severe diseases that can cause significant damage due to high mortalities of up to 70% in the Asian seabass aquaculture industry [3,4,5,6]. Therefore, this study aims to investigate the efficacy of nanovaccines derived from the inactivated bacteria *S. iniae* and *F. covae* in enhancing the immune response, gene expression related to immunity, and disease resistance against these two target pathogens in Asian seabass at the nursery farm scale. The information obtained from this study is expected to provide valuable information that can be applied to improve disease prevention in the Asian seabass aquaculture industry.

## 2. Materials and Methods

### 2.1. Fish and Experimental Designs

Healthy Asian seabass (*Lates calcarifer*) larvae (approximately 15 days after hatching (DAH)) with weights and lengths of 0.07 ± 0.02 g and 1.8 ± 0.17 cm, respectively, were raised at a nursery farm located in the Songklong subdistrict, Bangpakong district, Chachoengsao Province, Thailand. Fish larvae were raised and reared in 12 3.2 × 3.2 m^3^ cement ponds containing approximately 5000 L (50 cm in depth) of 5 ppt estuarine water with 50,000 fish/pond (10 fish/L stocking density). Four groups (3 ponds/group) in four different zones (1–4) for further vaccination experiments were constructed in nursery farm conditions. The water quality of the nursery pond was closely monitored and controlled daily to ensure its optimal levels for fish, with a temperature range of 28–30 °C, pH 7–8, salinity 3–5 ppt, and alkalinity 90–100 mg/L as CaCO_3_. The feeding schedule consisted of commercial pelleted feed twice daily, at 10% body weight, and the water was routinely changed every morning and evening, approximately 80%. Furthermore, routine fish sampling was conducted to test for potential diseases, including bacteria, viruses, and parasites, using standard microbiology methods [16].

All described experiments in the current study were conducted according to the Ethical Principles and Guidelines for the Use of Animals National Research Council of Thailand and approved by the Animal Ethics Committee, KU, Thailand (ACKU63-FIS-003).

### 2.2. Bacterial Culture and Formalin-Killed Vaccine Preparations

*Streptococcus iniae* (AQAHMSi2) and *Flavobacterium covae* (AQAHMFc) were isolated from moribund Asian seabass [17]. The bacterium was cultured in tryptic soy broth (TSB) (Difco™, Cockeysville, MD, USA) for *S. iniae* and Shieh’s broth for *F. covae*. These bacteria were incubated at 30 °C for 18–24 h (h). After incubation, the bacterial cultures were harvested via centrifugation at 5000 rpm for 10 min (min), and the resulting supernatant was discarded. The obtained pellets were washed twice with 0.85% NaCl. The bacterial cells were fixed with 1.0% formaldehyde solution for 24 h and washed twice with 0.85% NaCl. The bacterial cell density was measured using a spectrophotometer (Thermo Fisher Scientific, Waltham, MA, USA) by monitoring the optical density at 600 nm. The purity of the cultured bacterium was confirmed through microscopic examination (Olympus, Westborough, MA, USA) and basic Gram staining for further nanovaccine preparation.

### 2.3. Formulation of Bivalent Nanovaccines

The dry form of each nanovaccine of *S. iniae* and *F. covae* was prepared by complexing antigens with cationic biopolymers to form nanoparticles following the protocol of Kitiyodom et al. (2019) [18], with some modifications. Both *S. iniae* and *F. covae* were separately prepared and combined when used as bivalent vaccinations. To formulate the inactivated nanovaccine, both formalin-killed bacteria were adjusted based on an optical density until equivalent to 1 × 10^9^ CFU/mL. The bacterial cells were sonicated at 40% amplitude for 10 min in an ice bath to break them down into nanosized particles. Chitosan solution (cationic biopolymer) (50–200 kDa, Sigma, Saint Louis, MO, USA) at 0.25% *w*/*v* was prepared in 1% acetic acid.

To prepare the biopolymeric nanovaccine, sonicated bacterial cells were gently mixed with chitosan solution at a ratio of 1:0.5 (*v*/*v*). Chitosan was simultaneously complexed with antigen and formed cationic nanoparticles as mucoadhesive nanovaccines. The mixture was constantly stirred for 90 min at room temperature. Furthermore, the polymeric nanovaccine was transformed into dry form, treated with 2.5% sucrose, and then dried by a freeze-drying process. The vaccines were stored at −80 °C. The physicochemical properties of the nanovaccine were measured by average diameter and zeta potential using a Malvern Instruments Zetasizer Nano ZX, employing the dynamic light scattering (DLS) technique described by Kitiyodom et al. (2019) [18].

Before use, a dry form of bivalent nanovaccine of each *S. iniae* and *F. covae* nanovaccine was diluted with 0.85% NaCl to reach 5 × 10^8^ CFU/mL. Diluted nanovaccines of each bacterium were mixed at a ratio of 1:1, providing a 1 × 10^9^ CFU/mL mixture in a 2 L sterile bottle, and further used immediately for immersion vaccination after preparation. A preliminary analysis based on a mixture of *S. iniae* and *F. covae* nanovaccines showed that this ratio yielded higher ratios of specific IgM to *S. iniae* concentrations and specific IgM to *F. covae* concentrations compared to ratios of 1:2, 1:3, 2:1, and 3:1. This information was used for both oral and immersion vaccination.

For the oral vaccine, a nanovaccine of each *S. iniae* and *F. covae* with a concentration of approximately 8.5 × 10^6^ CFU was thoroughly mixed to provide a bivalent vaccine of the two bacteria of 1.7 × 10^7^ CFU. This concentration was used to mix with fish feed at 1.7 × 10^7^ CFU/fish in feed vaccination trials below.

### 2.4. Immersion Vaccination

Three ponds in zone 1 were used as a control, and three ponds in zone areas 2–4, which were maintained by different workers, were used for vaccine treatments. When the fish reached 30 DAH, the water in each nursing pond in zones 1–4 was reduced and remained 10 cm deep (1.0 m^3^ in volume). Then, 2 L of the above-prepared vaccines was premixed with 2 L of culture water and further thoroughly splashed in the nursing pond to reach the final concentrations of 2 × 10^6^ CFU/mL for both bacteria; fish larvae were exposed to vaccines for a full 20 min. Subsequently, similar spared water was added to a 50 cm depth to maintain a 4 × 10^5^ CFU/mL vaccine concentration. Fish larvae were exposed to the last concentration for 24 h. Two liters of 0.85% NaCl were added to the three control ponds, which served as the control group. Afterward, the water was replaced under the same conditions for 10 days. After this, the second immersion vaccination was conducted. Fish larvae at 40 DAH were booster-vaccinated under the same conditions as in the first immersion vaccination. During this period, aeration was fully supplied throughout the experiments.

### 2.5. Experimental Feed Preparation and Oral Vaccination

When the experimental fish reached 50 DAH, the total amount of a bivalent nanovaccine prepared in Section 2.3 was calculated based on the target concentration of 1.7 × 10^6^ CFU/fish and then mixed with 70 mL 0.85% NaCl/kg feed. The vaccine solution was thoroughly mixed with commercial feed (Thai Union Feedmill, Kalong, Thailand) at a 5% feeding rate and air-dried until completely dry. Afterward, experimental fish groups in zones 2–4 were fed 2 times with the prepared feed to reach a total vaccine concentration of 3.4 × 10^7^ CFU/fish/day. Fish were further fed experimental feed for 3 consecutive days to receive a net oral vaccine dose of approximately 1 × 10^8^ CFU/fish. For the control group, fish were fed with feed mixed with 0.85% NaCl under the same conditions as vaccinated fish.

### 2.6. Effects of Bivalent Nanovaccine on IgM and Immune Responses of Larvae to Fingerling Stages of Asian Seabass

#### 2.6.1. Fish Sampling

During the vaccination experiments, the whole body of Asian seabass was collected 10 days after each vaccination (10, 20, and 30 DAV or 40, 50, and 60 DAH, respectively). At each DAV, 24 fish were randomly selected. The first 8 fish were used for total IgM detection and IgM specific to *S. iniae* and *F. covae* by ELISA, lysozyme, and bactericidal activity. Another 8 fish were used for immune-related gene expression. The last 8 fish were prepared for histopathological changes in mucosal-associated lymphoid tissues (MALTs) in the experimental fish’s skin, gills, and gut.

#### 2.6.2. Extraction of Whole-Body Protein

At 10, 20, and 30 DAV, the whole body of Asian seabass was collected using sterilized scissors to finely cut fish for 1.5 mL Eppendorf tubes. Subsequently, sample tissues were preserved in a protein extraction buffer containing a protease inhibitor cocktail (HiMedia, Mumbai, India) at a ratio of 100 mg of tissue per 1 mL of protein extraction buffer. The Asian seabass samples were homogenized using pellet pestles in a stable temperature condition at 4 °C. After centrifugation at 2000–2500 rpm for 5 min at 4 °C, the supernatant was transferred to a fresh 1.5 mL Eppendorf tube and stored at −80 °C for ELISA and innate immune parameter analyses below.

#### 2.6.3. Total RNA Extraction and Preparation of First Strand cDNA

The whole fish samples were quickly preserved in 1.0 mL of TriZOL^TM^ reagent (Thermo Fisher Scientific, Waltham, MA, USA) according to the manufacturer’s protocol. Briefly, total RNA was subsequently isolated from the whole by immediately homogenizing fish samples in TriZOL^TM^ reagent with an automatic tissue extractor (MP, Irvine, CA, USA). Extraction steps were carefully carried out and the concentration was determined using a NanoDropTM spectrophotometer (Thermo Fisher Scientific, MA, USA). To synthesize first-strand cDNA, one microliter of 1000 ng/μL total RNA was used as a template with the ReverTra Ace^®^ qPCR RT Master Mix with gDNA Remover Kit (TOYOBO, Kita-ku, Osaka, Japan) according to the manufacturer’s protocol. The product of the first-strand cDNA synthesis was stored at −80 °C for further experiments.

#### 2.6.4. Histology Analysis of Mucosa-Associated Lymphoid Tissues (MALTs)

Whole fish bodies collected from all experimental groups after vaccination at each DAV were preserved and kept in Davidson’s fixative. The whole fish bodies were then embedded in paraffin blocks after being dehydrated and processed according to Fischer et al. (2008) [19]. Hematoxylin and eosin (H&E) were used to stain the target MALTs, including the skin, gills, and intestine. Histopathological changes were observed under a light microscope (Olympus, Littleton, MA, USA).

#### 2.6.5. Total Serum IgM

Total serum IgM protein from the whole body of experimental Asian seabass at 10, 20, and 30 DAV was used in the ELISA. The first step was to coat a flat-bottomed 96-well plate with bicarbonate/carbonate coating buffer, pH 9.6, with a volume of 50 mL, and incubate it at room temperature (RT) for 2 h and discard the solution. The microplate wells were then incubated overnight at 4 °C with 50 mL of fish serum protein. After incubation, the wells were washed three times with wash buffer containing phosphate-buffered saline (PBS) and 0.05% Tween-20 (PBST; pH 7.4). Then, 50 mL of VisualProtein-BlockPRO™ Blocking Buffer (Energenesis Biomedical Co., Ltd., Taipei, Taiwan) was added to each well and incubated for 2 h at RT. After incubation, the wells were washed thrice with wash buffer and incubated for 5 min at RT. After incubation, rabbit anti-Asian seabass IgM antibody (GenScript Biotech, Piscataway, NJ, USA) diluted to a ratio of 1:2000 was added to each well at a volume of 50 mL and incubated for 2 h at RT. After incubation, the wells were washed thrice with wash buffer using an ImmunoWash machine, set for 5 min for the final wash. Then, goat anti-rabbit IgM (Sigma, Springfield, MO, USA) diluted to a ratio of 1:5000 was added to a final volume of 50 mL and incubated at RT for 2 h. Afterward, wells were washed with wash buffer using the ImmunoWash machine 6 times and further incubated at RT for 5 min for the 6th wash step. The TMB substrate One Component HRP Microwell Substrate (Surmodics IVD, Inc., Eden Prairie, MN, USA) was added to a volume of 50 mL and incubated for approximately 1 min at RT, and then TMB Stop Solution (Surmodics IVD, Inc., USA) was added to stop the ELISA reaction. Finally, the IgM levels were measured from the absorbance of each reaction using the iMark™ Microplate Absorbance Reader at 450 nanometers (Bio-Rad Laboratories Ltd., Hercules, CA, USA).

#### 2.6.6. Assessment of Serum IgM Specific to *S. iniae* and *F. covae* Antigens

Detection of specific IgM against *S. iniae* and *F. covae* was performed using ELISA. Formalin killed the *S. iniae* and *F. covae* bacterial solutions at a 1 × 10^8^ CFU/mL concentration, which were prepared with the above protocol. Bacterial cells were sonicated with a VCX130 ultrasonic sonicator (Sonics & Materials, Inc., Newtown, CT, USA). The first step was to coat a flat-bottomed 96-well plate with bicarbonate/carbonate coating buffer, pH 9.6, with a volume of 50 mL, and incubate it at RT for 2 h before removing the solution. Next, the bacterial solution prepared earlier was added to each well with a volume of 50 mL and incubated overnight at 4 °C. Further steps for serum IgM specific to *S. iniae* and *F. covae* were conducted with the same conditions described in Section 2.6.5.

### 2.7. Humoral Innate Immune Response Assays

#### 2.7.1. Lysozyme Activity

The activity of serum lysozyme was evaluated in the lysis of *Micrococcus lysodeikticus* (Sigma, St. Louis, MO, USA), as previously described [20]. Lysozyme activity levels were determined using the following formula: units/mL enzyme = ((A540 sample − A540 blank) dilution factor)/(0.001) × (0.1).

#### 2.7.2. Bactericidal Activity

The bactericidal activity (BA) of the whole-body protein samples prepared as described in Section 2.6.2 was tested using *S. iniae* as the representative pathogenic bacterium in the bivalent nanovaccine formulation. The bacterium was cultured and prepared as described above. The BA of the experimental serum was determined by incubating 40 mL of protein serum with 10 μL of 1 × 10^5^ CFU/mL bacterial suspension of *S. iniae* bacterium in a 1.5 mL tube (total final pathogen cells in the reaction being 1 × 10^3^ CFU). The mixture was incubated for 2 h at RT. The surviving bacteria were counted on media specific to *S. iniae* [trypticase soy broth (Difco™ & BBL™, Cockeysville, MD, USA)]. Samples without serum and samples without bacteria were used as negative and positive controls (100% survival or 0% BA), respectively. The BA was calculated as the percentage of surviving bacteria after exposure to serum protein and present after plating on trypticase soy agar (Merck KGaA, Darmstadt, Germany) using the following formula: BA (%) = ((T_0_ − T_24_)/T_0_) × 100, where T_0_ is the total initial bacteria and T_24_ is the number of bacteria present after 24 h of exposure.

### 2.8. Expression of Immune-Related Genes of Larvae in Fingerling Stages of Asian Seabass Using Quantitative Real-Time PCR (qRT-PCR)

The immune-related genes *IgM*, *IgT*, *IgD*, *CD4*, *MHCIIα*, and *TCRα* were selected as targets in the expression analysis. *IgM* and *IgT* primers were specifically designed based on their secreted Ig forms. All the primers used in the study were validated in Asian seabass by RT-PCR amplification and nucleotide sequencing and are listed in Table 1.

The qRT-PCR assays were performed using Brilliant III Ultra-Fast SYBR^®^ Green (Agilent, Santa Clara, CA, USA) in an Mx3005P QPCR Systems instrument (Agilent, Santa Clara, CA, USA). The qPCRs were optimized in 15 μL reactions, including 10 μL of 2×SYBR Green QPCR Master Mix, 1 μL of 0.5 mM forward and reverse primers, 1 μL of cDNA template, and distilled water to adjust the reaction to a final volume of 15 μL. The qPCR cycling conditions consisted of an initial condition of one cycle of 95 °C for 5 min, 40 cycles of 95 °C for 30 s, 60 °C for 30 s, and 72 °C for 90 s, and a final extension at 72 °C for 10 min. Triplicate qRT-PCRs were conducted for each sample. The housekeeping gene *β-actin* was used to standardize the results by eliminating variation in mRNA and cDNA quantity and quality. The relative expression of immune-related genes in the whole body of Asian seabass at different time points was calculated using 2^−ΔΔCT^ analysis following the protocol of Livak and Schmittgen (2001) [21].

### 2.9. Growth Performance

The effects of bivalent nanovaccines on growth performance were assessed during a 30-day vaccination trial. The growth performances of all treatment groups were based on their body weight data. Thirty fish from each treatment group (10 fish/pond) were randomly monitored for growth performance parameters by weighing their bodies at 10, 20, and 30 DAV. Growth performances were reported as 1) absolute growth rate (AGR), including weight gain (WG) and average daily growth rate (ADG), and 2) specific growth rate (SGR). Moreover, the feed conversion ratio (FCR) was also measured in all experimental trials. All growth calculations were performed using the methods described by Bunnoy et al. (2019) [22].
WG (g or cm/30 days) = W_t_ − W_i_(1)
ADG (g or cm/day) = (W_t_ − W_i_)/t (2)
SGR (%/day) = log (W_t_) − log (W_i_)/t × 100 (3)
FCR = amount of total feed given/WG(4)
where W_t_ is the final weight/length, W_i_ is the initial weight/length, and t is the trial duration (30 days).

### 2.10. Challenge with S. iniae and F. covae

After 30 DAV, one hundred eighty fish from each pond of zones 1–4 were randomly selected and transferred into six 250 L fiberglass tanks (30 fish/tank) for challenge tests: three tanks for *S. iniae* and three tanks for *F. covae*. The preliminary pathogenicity of virulent *S. iniae* and *F. covae* on Asian seabass was determined to validate the optimum dose for the experimental challenge trials. The fourteen-day median lethal concentration (LC_50_) was assessed by direct immersion with the obtained final concentrations of 1 × 10^5^ CFU/mL for *F. covae* and 1 × 10^7^ CFU/mL for *S. iniae* for 30 min (data not presented), which was subsequently applied to the challenge test in this trial. The mortality and survival of the fish were recorded every 12 h for up to 14 days post-challenge. Re-isolation of *S. iniae* and *F. covae* from moribund fish was performed to verify the cause of death in fish using a standard diagnostic protocol [16]. Cumulative survival analysis of Asian seabass challenged with *S. iniae* and *F. covae* was performed using the Kaplan–Meier method [15]. The level of statistical significance between the control and experimental groups in the challenge test was indicated as * (*p* < 0.05) using Student’s *t*-test. The relative percent survival (RPS) of each vaccinated group was calculated based on the previous study [13].

### 2.11. Statistical Analysis

Protein serum IgM levels, innate immune response parameters, immune-related gene expression, and cumulative survival were obtained from three replicates, and the results are expressed as means ± standard deviations (SDs). Data were statistically analyzed by one-way analysis of variance (ANOVA) and Duncan’s new multiple range test (DMRT) to determine differences among groups. Significant differences among groups were statistically considered when *p* < 0.05–*p* < 0.001. All statistical analyses were performed using Statistical Package for Social Science (SPSS for Windows version 24.0, Chicago, IL, USA).

## 3. Results

### 3.1. Production and Characterization of S. iniae and F. covae Nanovaccines

In this part, *S. iniae* and *F. covae* nanovaccines were successfully developed for the first time. The physiological properties of these nanovaccines are indicated by average diameter and zeta potential values. As shown in Table 2, the zeta potential of both nanovaccines shifted from a negative charge for the sonicated antigen form to a positive charge for polymeric nanovaccines. It also confirmed the positive charge character of the cationic nanovaccines in both the *S. iniae* and *F. covae* vaccines. Moreover, both vaccines still obtained this characteristic after transformation into the dry form. The results also showed that the average diameters of polymeric nanovaccines (dry form) were slightly greater than those of the solution form in both *S. iniae* and *F. covae* vaccines.

### 3.2. Effects of Bivalent Nanovaccine via Immersion and Oral Vaccination on IgM and Immune Responses of Larvae to Fingerling Stages of Asian Seabass

#### 3.2.1. Total Serum IgM

At 10 DAV via immersion, a significant increase in total IgM levels indicated by the obtained absorbance was observed in fish of groups 3 and 4 with 0.475 ± 0.04 and 0.725 ± 0.16, respectively, compared to the control group with 0.294 ± 0.05 (*p* < 0.05). However, group 2 did not show a significant difference in total IgM levels compared to the control group (*p* > 0.05). Following 20 DAV via immersion, only group 4 exhibited a significantly higher total IgM level (0.782 ± 0.20) than the control group (0.444 ± 0.28) (*p* < 0.05). There was no significant difference in total IgM levels for vaccinated fish in groups 2 and 3 compared to the control group (*p* > 0.05). At 30 DAV via oral administration, both groups 3 and 4 exhibited significantly increased total IgM levels of 0.794 ± 0.17 and 0.973 ± 0.28, respectively, compared to the control group (*p* < 0.05). However, group 2 showed no significant difference in total IgM levels compared to the control group (*p* > 0.05) (Figure 1A).

#### 3.2.2. Serum IgM specific for *S. iniae* and *F. covae*

A significant increase in the protein serum IgM levels specific to *S. iniae* was observed in all groups immunized with the bivalent nanovaccine compared to the control groups at every DAV (Figure 1B). In the same way, a significant increase in the protein serum IgM levels specific to *F. covae* was mainly observed in all groups immunized with the bivalent nanovaccine compared to the control groups (Figure 1C).

At 10 DAV, the Asian seabass in groups 2, 3, and 4 exhibited significantly higher levels of specific IgM against *S. iniae* than those in the control group (*p* < 0.05). The specific IgM levels in groups 2, 3, and 4 were measured with absorbances of 0.918 ± 0.20, 0.827 ± 0.14, and 0.793 ± 0.38, respectively, while the control group had an absorbance of 0.440 ± 0.10. At 20 DAV, the fish in groups 2, 3, and 4 continued demonstrating significantly higher levels of specific IgM against *S. iniae* compared to the control group (*p* < 0.05), with absorbances of 1.074 ± 0.30, 1.094 ± 0.31, and 1.447 ± 0.32, respectively, while the control group had an absorbance of 0.514 ± 0.10. Similarly, at 30 DAV, the fish in groups 2, 3, and 4 maintained significantly higher levels of specific IgM against *S. iniae* (*p* < 0.05). The specific IgM levels in groups 2, 3, and 4 were measured with absorbances of 1.086 ± 0.08, 1.099 ± 0.18, and 1.449 ± 0.23, respectively, while the control group had an absorbance of 0.638 ± 0.09 (Figure 1B).

At 10 DAV, a significant increase in specific IgM levels against *F. covae* was observed in fish groups 2 and 3 compared to the control group (*p* < 0.05). The specific IgM levels in groups 2 and 3 were measured with absorbances of 0.462 ± 0.12 and 0.448 ± 0.10, respectively, while the control group had an absorbance of 0.282 ± 0.07. However, no significant difference was observed in the specific IgM levels against *F. covae* in group 4 compared to the control group (*p* > 0.05), with an absorbance of 0.336 ± 0.09. At 20 DAV, the fish in groups 2, 3, and 4 continued to exhibit significantly higher levels of specific IgM against *F. covae* than those in the control group (*p* < 0.05). The specific IgM levels in groups 2, 3, and 4 showed absorbances of 0.636 ± 0.15, 0.744 ± 0.18, and 0.840 ± 0.15, respectively, while the control group had an absorbance of 0.402 ± 0.09. Similarly, at 30 DAV, the fish in groups 2, 3, and 4 exhibited significantly higher levels of specific IgM against *F. covae* than those in the control group (*p* < 0.05). The specific IgM levels in groups 2, 3, and 4 indicated absorbances of 0.860 ± 0.15, 0.845 ± 0.18, and 1.102 ± 0.25, respectively, while that in the control group was 0.477 ± 0.10 (Figure 1C).

#### 3.2.3. Nonspecific Humoral Immune Responses

##### Bactericidal Activity (BA)

At 10 and 20 DAV, a significant increase in bactericidal activity against *S. iniae* was observed only in Asian seabass that were immunized with the bivalent nanovaccine in group 4 (*p* < 0.05), with 57.21 ± 8.86% and 64.22 ± 6.21%, respectively. Interestingly, at 30 DAV, a significant increase in the BA against *S. iniae* was observed in all groups immunized with bivalent nanovaccines, with 61.03 ± 9.21%, 66.166 ± 6.96%, and 79.04 ± 6.66% for groups 2, 3, and 4 (*p* < 0.05), respectively (Figure 2A).

##### Lysozyme Activity

The lysozyme activity in the protein serum of all vaccinated groups increased rapidly at 10 DAV after the first vaccination, reaching 88.71 ± 33.67, 157.09 ± 29.45, and 168.96 ± 27.00 unit/mL for groups 2, 3, and 4, respectively, and still significantly differed at 20 DAV and 30 DAV, which exceeded the levels of the control group by a substantial margin (*p* < 0.05). In comparison, the protein serum lysozyme activity of the control group fluctuated slightly from 10 DAV to 30 DAV, which was not a significant change (*p* > 0.05) (Figure 2B).

### 3.3. Effects of Bivalent Nanovaccines on the Expression of Immune-Related Genes

The relative expression levels of immune-related genes, including *IgM*, *IgT*, *IgD*, *MHCIIα*, *TCRα*, and *CD4* in the whole body are presented in Figure 3A–F. At 10 DAV, the expression of the *IgM* gene was significantly upregulated in all vaccinated fish groups by 2.63 ± 0.81-, 1.572 ± 0.48-, and 2.03 ± 0.58-fold, respectively, compared to the control group (*p* < 0.05) (Figure 3A). Similar results were observed at 20 DAV, when all vaccinated fish showed significant upregulation of *IgM* expression levels (*p* < 0.05) compared to the control (Figure 3A). However, at 30 DAV, a significant difference in *IgM* compared to the control was observed only in group 4 (Figure 3A). The expression patterns of the *IgT* gene were found to be similarly significantly upregulated at 10 and 20 DAV. At 10 DAV, vaccinated groups 3 and 4 demonstrated highly significant differences in *IgT* greater than the control group by 74.68 ± 14.08- and 56.92 ± 16.64-fold, respectively (*p* < 0.001) (Figure 3B). At 20 DAV, these two groups still showed 83.33 ± 19.25- and 60.13 ± 13.97-fold upregulation, respectively, compared to the control group (*p* < 0.001) (Figure 3B). At 30 DAV, the *IgT* gene expression in all vaccinated fish groups was also upregulated significantly by 7.43 ± 4.44-, 72.66 ± 10.17-, and 61.89 ± 4.45-fold for groups 2, 3, and 4, respectively (*p* < 0.05 and 0.001) (Figure 3B). The expression of *MHCIIα* observed at 20 DAV in all vaccinated fish groups 2, 3, and 4 recorded 2.22 ± 0.73-, 2.73 ± 0.69-, and 2.74 ± 0.79-fold changes, respectively, compared to the control group (*p* < 0.05) (Figure 3D). The expression of *TCRα* was also observed at 10 DAV in vaccinated fish groups 2 and 3 to exhibit 1.14 ± 0.32- and 1.29 ± 0.34-fold changes, respectively. On day 20, *TCRα* expression was upregulated in all vaccinated fish groups (groups 2, 3, and 4) by 1.41 ± 0.19-, 2.07 ± 0.42-, and 2.17 ± 0.49-fold, respectively, compared to the control group (*p* < 0.05) (Figure 3E). Regarding the *IgD* and *CD4* genes, no significant differences were observed in most vaccinated fish groups at 10, 20, and 30 DAV when compared to the control group (*p* > 0.05) (Figure 3C,F).

### 3.4. Effects of Bivalent Nanovaccine on Histopathological Changes in MALTs

At each tested period after vaccination, eight fish of each group were used to analyze the histological alteration in MALTs. Interestingly, at every tested time point after vaccination, the fish in all the vaccinated groups exhibited a similar histology, which was different from that of the control fish.

#### 3.4.1. SALT

Histological analysis of SALT in Asian seabass following bivalent nanovaccination showed widened lateral line pores after the first, second, and third vaccinations at 10, 20, and 30 DAV, respectively. Fish in all vaccinated groups had significantly wider lateral line tubules compared to the control group (Figure 4). Additionally, after the first and second vaccinations, a notable presence of blue-stained cells (immune-like or macrophage-like cells) was observed around the lateral line tubules in the vaccinated group compared to the control group. Moreover, on the dermis and epidermal layers of the vaccinated fish after the first, second, and third vaccinations, a substantial number of immune-like cells were found in the respective areas compared to the control group (Figure 5).

#### 3.4.2. GIALT

At 10, 20, and 30 DAV, the histological changes in the fish gills revealed no significant differences in the number of goblet cells (Figure 6). However, it was observed that the epithelial cells of the gill lamellae in vaccinated fish appeared enlarged when compared to the control group (Figure 6). Moreover, in the interbranchial lymphoid tissue (ILT), which is part of the GIALT located on the gills, there were no significant changes after the first vaccination (10 DAV). Interestingly, however, after the second and third vaccinations, the ILT showed the noticeable expansion and infiltration of white blood cells (Figure 7).

#### 3.4.3. GALT

The results revealed no abnormal changes in the overall structure of the intestine or epithelial cells at 10 and 20 DAV, indicating the absence of significant histopathological alterations in the cells and tissues. No clinical signs or histopathological lesions were observed in the fish that received the nanovaccines compared to the control group. However, a slight increase in the number of goblet cells responsible for mucus production was noted in both the vaccinated and unvaccinated fish. Nevertheless, no significant differences were found compared to the control group (Figure 8A–L).

At 30 DAV, the fish intestine’s overall structure, epithelial cells, and goblet cells displayed no significant histopathological changes. However, a notable difference was observed in the number of goblet cells in all vaccinated groups, which exhibited a significant increase in these cells compared to the control group. Additionally, the intestine of vaccinated fish showed a thicker laminar propria layer than that of the control group (Figure 8P–R).

### 3.5. Growth Performance

The effect of the bivalent nanovaccines on the growth performance of vaccinated Asian seabass is shown in Table 3. After 30 DAV, no significant differences in growth parameters were measured, i.e., WG, ADG, SGR, and FCR, between the control and the treatment groups (*p* > 0.05).

### 3.6. Challenge Tests with S. iniae and F. covae

The survival rates of Asian seabass immunized with bivalent nanovaccines in all groups, including groups 2, 3, and 4, were 62.15 ± 2.11, 60.00 ± 3.00, and 75.70 ± 3.36%, respectively, (with the RPS of 31.92, 28.81, and 56.29, respectively) after challenge with *S. iniae*. These values were significantly higher than those of the control group (*p* < 0.05), with a cumulative survival rate of 44.44 ± 1.92% (Figure 9A). Similarly, the Asian seabass immunized with bivalent nanovaccines in all groups, including groups 2, 3, and 4, showed cumulative survival rates of 68.89 ± 3.85, 64.44 ± 5.09, and 77.78 ± 5.09%, respectively, (with the RPS of 37.37, 28.41, and 55.26, respectively) after challenging the fish with *F. covae*, which were significantly higher than that of the control group (*p* < 0.05), with a cumulative survival of 50.00 ± 3.33% (Figure 9B).

## 4. Discussion

Intensive Asian seabass cultures are constantly faced with an increase in the risk of disease outbreaks, leading to a reduction in fish production and mass mortalities [23]. Streptococcosis and columnaris disease are common causes of disease outbreaks in many freshwater fish species worldwide, mainly in the Asian seabass culture in Thailand [5,24]. The efficacy of antibiotics and chemicals in managing these diseases is variable. Vaccination offers a pragmatic and trustworthy means of averting diseases in aquaculture, diminishing occurrences of mass mortality, and lessening the reliance on harmful antibiotics and chemicals [25,26]. These strategies have been commercially applied to many economic fish species [27]. In comparison to the vaccination routes, injection administration is the most effective regarding immune responses and protection levels [8]. However, this method is relatively unacceptable for commercial scales with high labor, high stress, mortality induction, booster vaccination difficulty during fish growth in grow-out ponds, cost requirements, etc. Therefore, immersion and oral vaccination have been developed to increase their efficacy in the fish aquaculture industry. These methods were previously found to exhibit low efficacy in protection but lately have been developed to meet an acceptable level important for industry. With the progress and development of nanotechnology, nanovaccines with mono- and bivalent characteristics have been effectively developed [12,28].

Therefore, the development of bivalent nanovaccines against *S. iniae* and *F. covae* was conducted in the present study to generate high efficacy in resistance to these two harmful pathogenic bacteria in Asian seabass larvae.

Based on the physical characteristics of *S. iniae* and *F. covae* nanovaccines, it was demonstrated that the average particle size and zeta potential were in line with the suitable and influential characteristics of nanovaccines that are smaller than 500 nm and have positive charges. The cationic biopolymer generated positively charged nanovaccines, enhancing adhesion on the mucosal surface and improving protection against diseases. These properties have been suggested as the optimal characteristics for effective and favorable vaccines to be efficiently absorbed by fish immune-associated organs or tissues [29].

Recently, mono- and bivalent nanovaccines have been developed to provide better immune responses and protection in various fish species, especially tilapia, from several fish pathogens, such as *Aeromonas veronii* [30], *Flavobacterium oreochromis* [12], *Francisella orientalis* [14], *Streptococcus agalactiae* [31,32], and tilapia lake virus [33,34,35]. These nanovaccines were formulated from various materials, including chitosan or its derivatives, alginate, nanoclay, halloysite nanotubes, and poly[(methyl methacrylate)-co-(methyl acrylate)-co-(methacrylic acid)]-poly(d,l-lactide-co-glycolide) (PMMMA-PLGA). With oral, immersion, and intramuscular injections, these nanovaccines effectively elevated protection from the pathogens mentioned above [36]. Of these, chitosan and its derivatives are low-cost and effective adjuvants, enhancing better immune responses such as immunostimulants [37,38].

To date, various monovalent nanovaccines have been reported in some fish species; in the present study, chitosan-based bivalent nanovaccines created from *S. iniae* and *F. covae* bacteria were further applied to fish larvae in the nursery farms of farmers via the first immersion and the other two immersions during routine water exchanges of larval stages and oral booster vaccination during fingerling periods, respectively. These combination methods have been reported to prevent stress conditions that may severely rise during or after excessive handling or overcrowding conditions in vaccination periods [39,40]. Our research intends to explore cost-effective, safe, practical, and high-efficacy vaccines for dually preventing harmful pathogenic bacteria causing streptococcosis and columnaris diseases at Asian seabass farm scales.

Principally, total serum IgM is expected to be the first indicator used to demonstrate immune responses after vaccination, and nanovaccines effectively surpassed this target parameter [13]. The application of the current nanovaccines significantly increased serum IgM levels in the 3rd and 4th vaccinated groups at 10 and 30 DAVs, similar to bivalent mucoadhesive nanovaccines of *Flavobacterium oreochromis* (For) and *Francisella orientalis* (Fo) and monovalent nanovaccines of For and Fo, which showed significant IgM levels compared to the control group. However, when specific IgM to *S. iniae* or *F. covae* bacteria was measured, all nanovaccine-immunized groups showed significantly higher specific IgM than the unvaccinated control group at every DAV. In contrast to the previous study, the total IgM levels of Nile tilapia in bivalent mucoadhesive nanovaccine *For*- and *Fo*-vaccinated groups were significantly suppressed compared to the *For* or *Fo* monovalent vaccines [12], suggesting that the bivalent *S. iniae* or *F. covae* nanovaccine is strong enough to enhance specific IgM levels against both *S. iniae* or *F. covae*.

Additionally, our findings showed that fish immunized with bivalent nanovaccines showed an increase in innate immune parameters, including lysozyme activity and bactericidal activity. This information agreed with the results from previous studies conducted by Bunnoy et al. [12], which showed that the application of bivalent mucoadhesive nanovaccines to prevent francisellosis and columnaris diseases in Nile tilapia also significantly elevated these innate immune parameters against *Francisella orientalis* and *Flavobacterium oreochromis*, suggesting that the innate immune responses of fish are simultaneously induced by bivalent nanovaccine application. Furthermore, this vaccine effectively upregulates several immune-related genes, such as *IgM*, *IgD*, *IgT*, *MHCIIα*, *TCRα*, and *CD4*, which are key components of specific immune systems. This information is firmly in line with previous studies [12,14,28,41,42,43], suggesting that the bivalent nanovaccines in the current study are potent immunogens that vigorously drive the two arms of both the innate and adaptive immune systems [12]. Particularly for *IgM* and *IgT*, almost all vaccinated treatments showed very high upregulation, indicating that two immersion and one booster vaccination of the bivalent nanovaccines effectively enhance both systemic and local specific immune responses. Similar responses were also observed in several monovalent nanovaccine experiments [14,18,30] and a bivalent nanovaccine [12].

Histopathology intensively supported the surpassing of local immune responses found in the above information. Our study demonstrated that bivalent nanovaccines could permeate the skin and gills, the main organs targeted by the immersion vaccination method [44]. Interestingly, the interbranchial lymphoid tissue (ILT) in the gills of all vaccinated fish became elevated, with numerous immune-like cells related to the immune responses. However, this structure, the ILT, does not seem to have any equivalent among lymphoid tissues, and its function is still unknown, although it shares some properties with secondary lymphoid structures [45]. Additionally, an elevation of the epithelium layer was observed in the gills of the vaccinated fish. These changes have never been reported previously in immersion vaccination trials, but similar characteristics can be observed in previous experiments, where fish were exposed to various factors [46,47,48]. This mysterious observation requires further investigation.

Furthermore, histopathological changes in mucus cells on epithelial layers were also observed in the gut of all vaccinated fish after the second booster vaccination via oral administration at 30 DAV. This phenomenon strongly indicates that an effective oral vaccination route impacts immune responses in GALT, which is crucial for the local gastrointestinal tract resistant components of fish [49,50,51]. Unfortunately, no available studies dealing with vaccination in fish using bivalent nanovaccines can be found for comparative analysis.

For the current study, research on combined nanovaccines using Gram-positive and Gram-negative bacteria for Asian seabass or other fish species is still scarce. Therefore, there is insufficient data to clearly analyze and compare the mechanism of combined nanovaccine responses in Asian seabass in the current study. Further research is needed to address these gaps in the future. However, previous studies on other fish species, especially tilapia, have demonstrated positive results, indicating the significant impacts of nanovaccine application with various practical administrations.

Recently, the efficacy of the whole-cell-based monovalent and bivalent vaccines of *S. iniae* and *F. covae* in fingerling Asian seabass has been investigated via both oral and injection vaccination by our research group [17]. It was found that bivalent vaccines of these two bacteria had poorly elevated immune responses and prevention against these two bacteria compared to each monovalent vaccine. In our current results, chitosan-based bivalent nanovaccines showed survival rates of 60.00–75.70% for *S. iniae* challenge and 64.44–77.78% for *F. covae* testing. This information is similar to previous reports that the high efficacy of chitosan-based nanovaccines indicated by the RPS or survival rate was found in various experiments. For example, chitosan-based monovalent nanovaccines against *Aeromonas veronii* [30], *Flavobacterium columnare* [13,18], *Streptococcus agalactiae* [31], and tilapia lake virus [34] immunized with oral or immersion vaccinations showed a high RPS or survival ranging from 52.2–100%. This result suggested that the chitosan-based bivalent nanovaccines, which were immunized via the combination of immersion and oral administration in Asian seabass larvae to fingerlings, led to an additive action, dually protecting Asian seabass from columnaris and streptococcosis effectively.

Considerably, some immune parameters of vaccinated groups 2–4 differed at some periods, which may be effectively due to different management methods of workers and should be optimized and considered when establishing a vaccination strategy for nursery farms.

## 5. Conclusions

In the current study, developing a novel and innovative bivalent nanovaccine can potentially become one of the most effective methods for controlling pathogenic *S. iniae* and *F. covae* in Asian seabass. This nanovaccine enhances the efficacy of vaccination against streptococcosis and columnaris disease by boosting both innate and adaptive immunity, as indicated by innate immune parameters, elevated protein serum antibody levels of both total and specific IgM, upregulated gene expression related to immunity, and improved survival rates and disease resistance. The combined suitable and practical routes of immersion and oral vaccinations are less labor-intensive, cost-effective, safe, practical, and high-efficiency, and provide dual protection against streptococcosis and columnaris diseases at the nursery Asian seabass farm scale, which are further crucial key roles and stepping stones for developing and sustaining the Asian seabass aquaculture industry worldwide.

## Figures and Tables

**Figure 1 vaccines-12-00017-f001:**
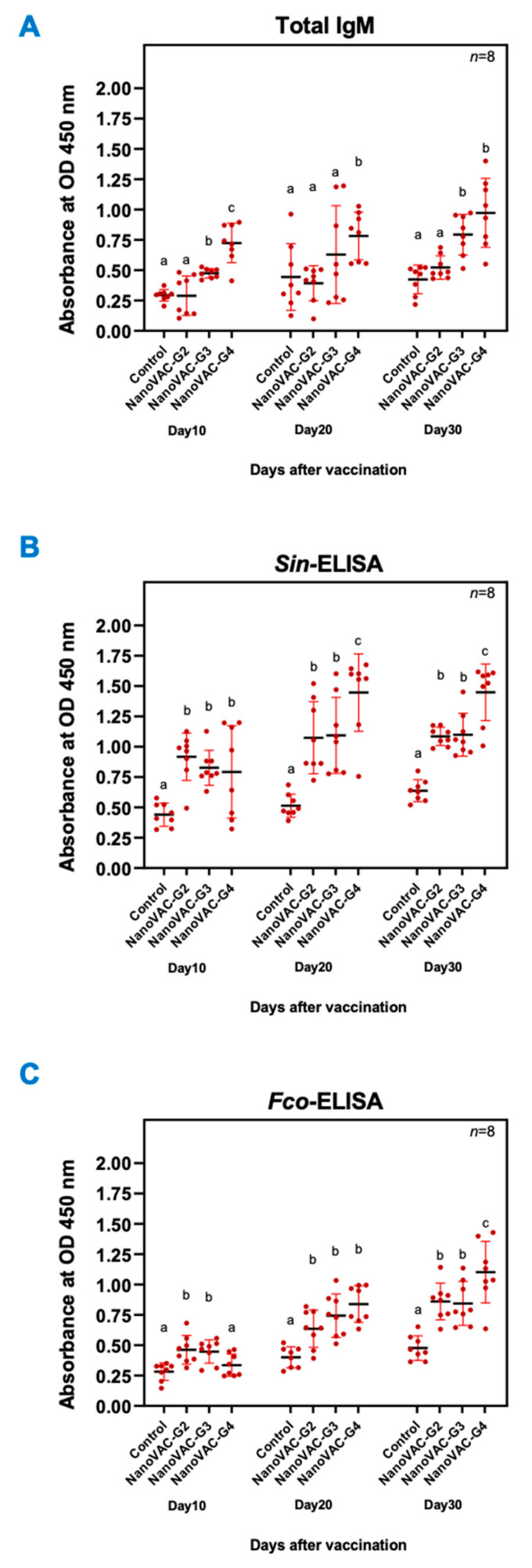
Total IgM (**A**) and IgM specific to *S. iniae* (**B**) and *F. covae* (**C**) in Asian seabass vaccinated with bivalent nanovaccines at 10, 20, and 30 DAV. All values are presented as means ± SDs (*n* = 8). Superscripted letters indicate differences among the treatment groups (*p* < 0.05).

**Figure 2 vaccines-12-00017-f002:**
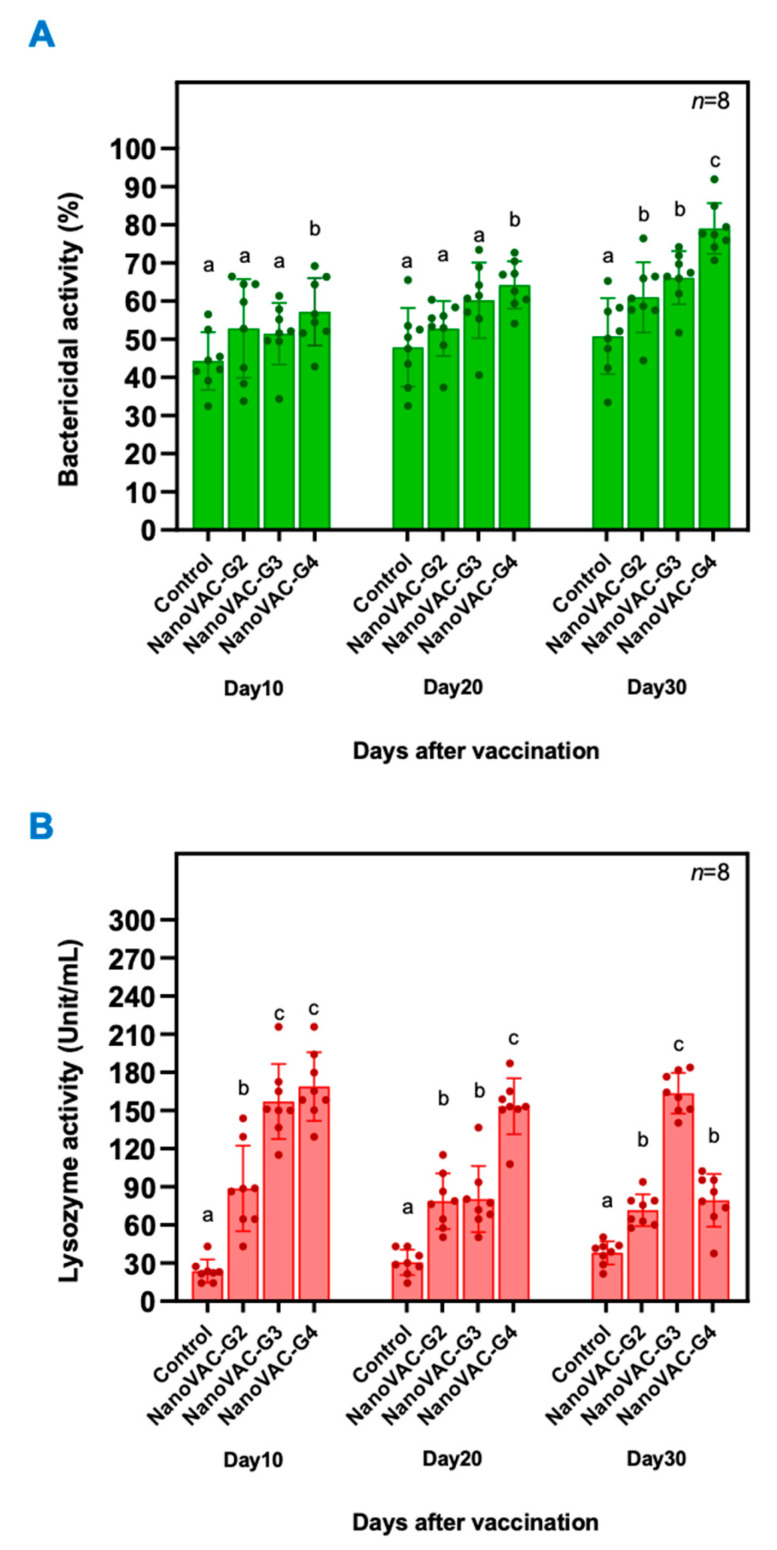
Bactericidal activity (**A**) and lysozyme activity (**B**) of Asian seabass vaccinated with bivalent nanovaccines at 10, 20, and 30 DAV. All values are presented as means ± SDs (*n* = 8). Superscripted letters indicate differences among the treatment groups (*p* < 0.05).

**Figure 3 vaccines-12-00017-f003:**
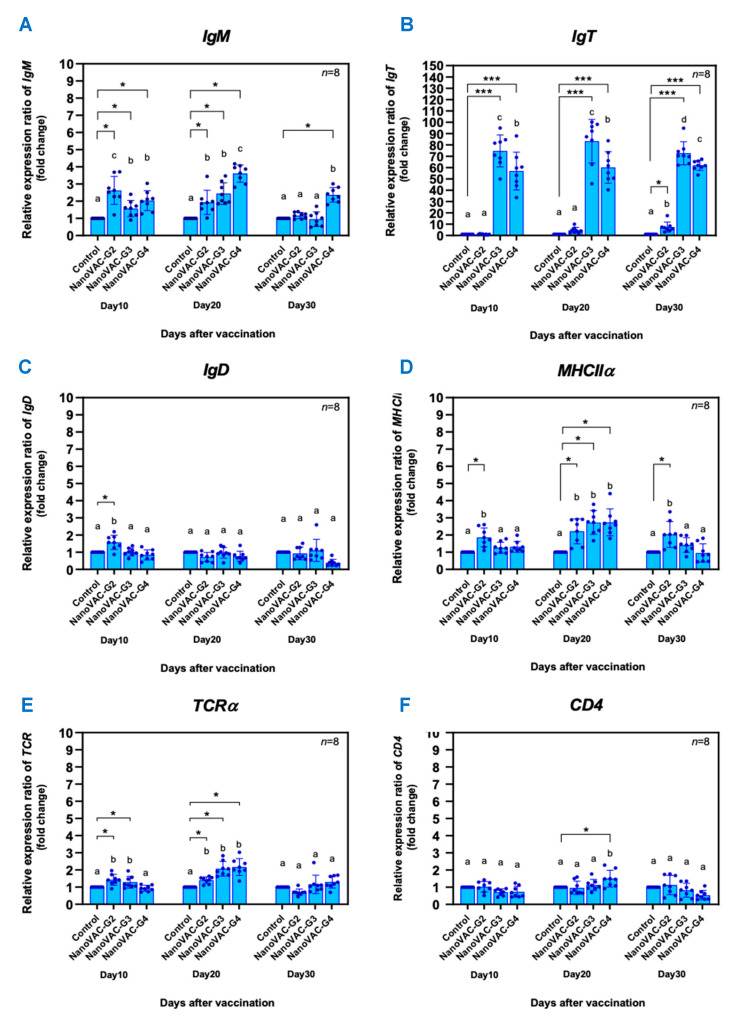
Relative gene expression levels of specific immune-related genes, including *IgM* (**A**), *IgT* (**B**), *IgD* (**C**), *MHCIIα* (**D**), *TCRα* (**E**), and *CD4* (**F**) in the whole body of Asian seabass vaccinated with bivalent nanovaccines at 10, 20, and 30 DAV. All values are presented as means ± SDs (*n* = 8). Superscripted letters indicate differences among the treatment groups (*p* < 0.05). Significant levels at *p* < 0.05 or *p* < 0.001 are indicated by * or ***, respectively.

**Figure 4 vaccines-12-00017-f004:**
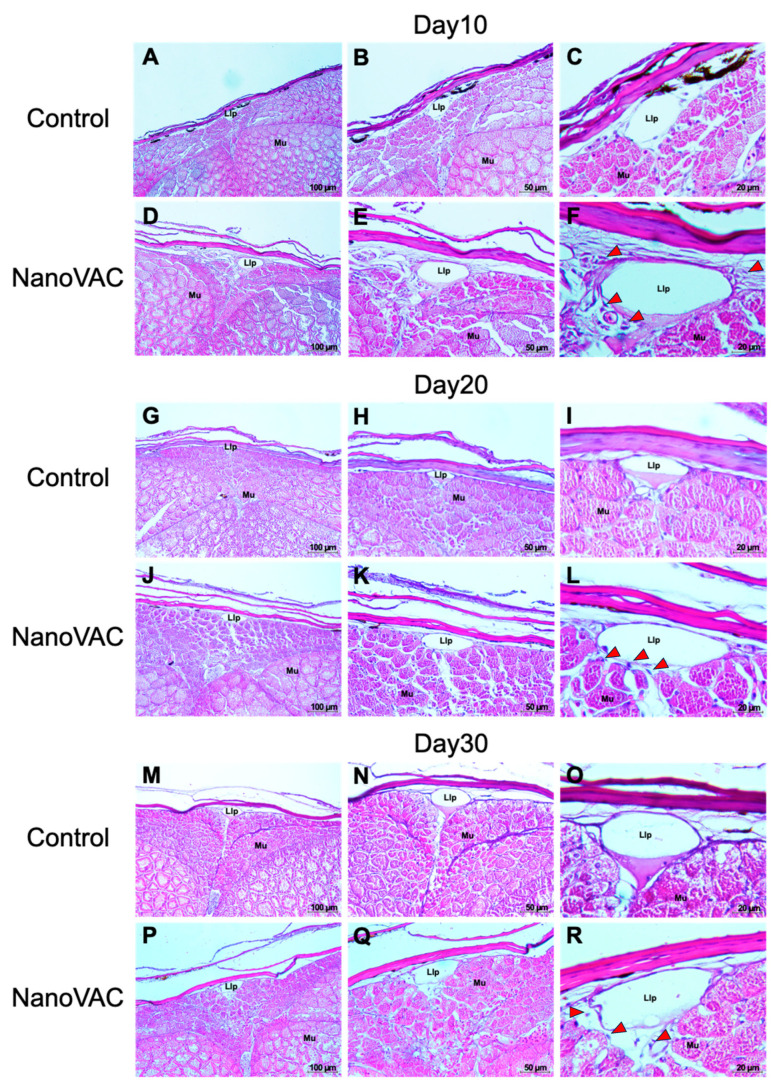
Histopathological changes in the skin of Asian seabass after immunization with bivalent nanovaccines at 10, 20, and 30 DAV. Llp: lateral line tubules; Mu: muscle (H&E staining: 20× (**A**–**P**), 40× (**B**–**Q**), and 100× (**C**–**R**) magnification). Red arrows indicate macrophage-like cells.

**Figure 5 vaccines-12-00017-f005:**
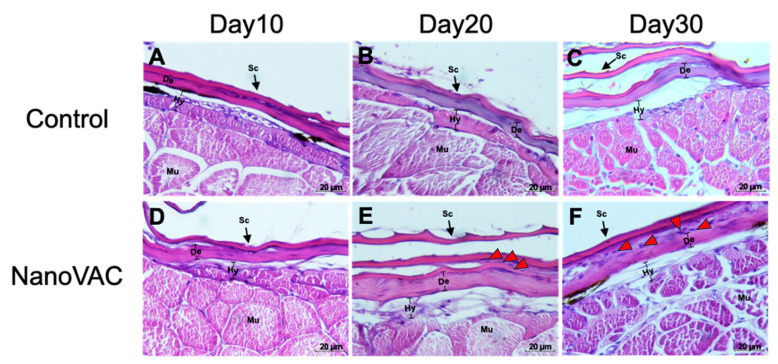
Histopathological changes in the skin of Asian seabass after immunization with bivalent nanovaccines at 10, 20, and 30 DAV (**A**–**F**). Sc: scale; Hy: hypodermis layer; De: dermis; Mu: muscle (H&E staining: 100× magnification). Red arrows indicate macrophage-like cells.

**Figure 6 vaccines-12-00017-f006:**
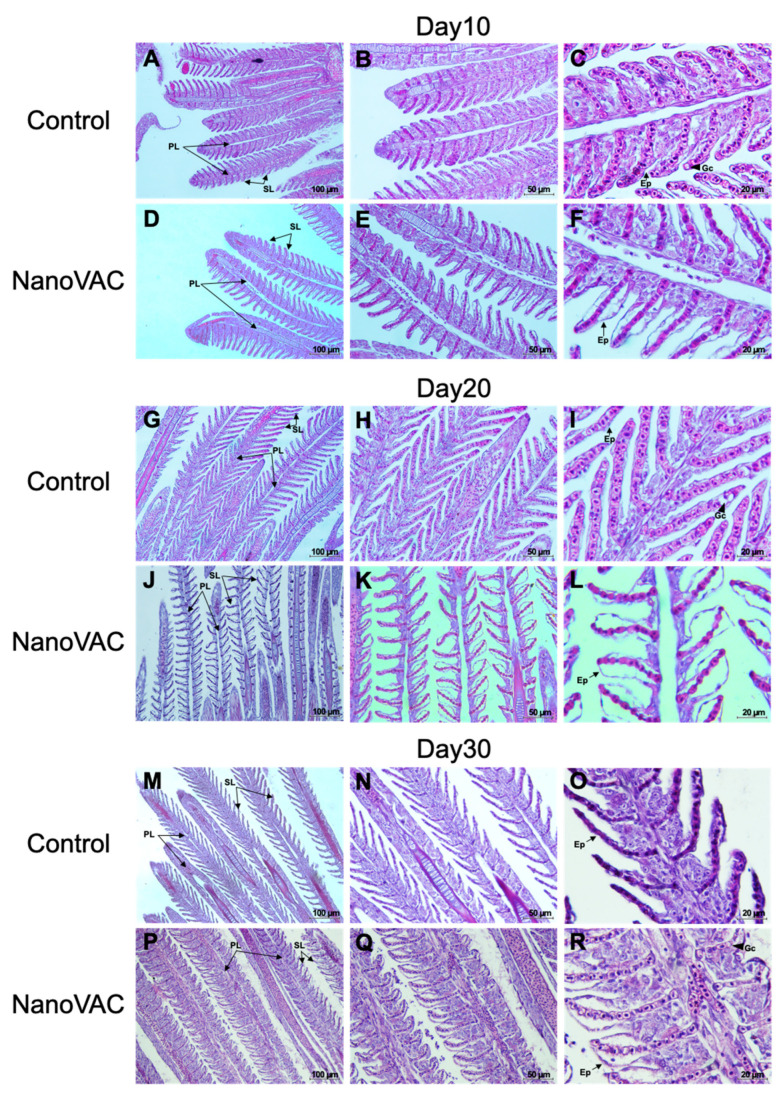
Histopathological changes in the gills of Asian seabass after immunization with bivalent nanovaccines at 10, 20, and 30 DAV. PL: primary lamella; SL: secondary lamella; Ep: epithelium; Gc: goblet cells (H&E staining: 20× (**A**–**P**), 40× (**B**–**Q**), and 100× (**C**–**R**) magnification).

**Figure 7 vaccines-12-00017-f007:**
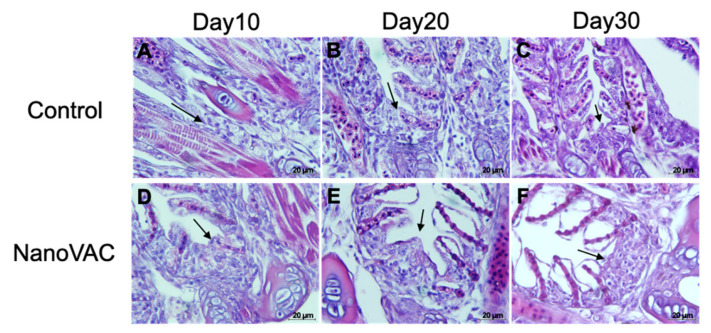
Histopathological changes in the gills, focusing on interbranchial lymphoid tissues (ILTs) of the control (**A**–**C**) and vaccinated (**D**–**F**) Asian seabass after immunization with bivalent nanovaccines at 10, 20, and 30 DAV. Black arrow: enlarged and cumulative immune-like cells of the ILTs (H&E staining: 100× magnification).

**Figure 8 vaccines-12-00017-f008:**
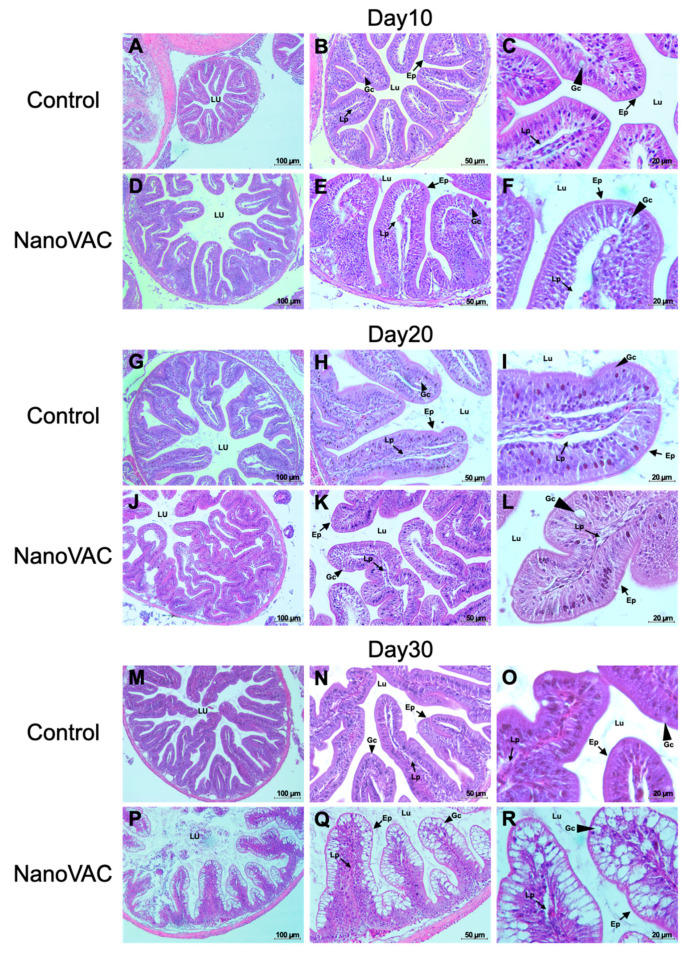
Histopathological changes in the gut of Asian seabass after immunization with bivalent nanovaccines at 10, 20, and 30 DAV. LU: lumen; Ep: epithelium; Lp: laminar propria; Gc: goblet cells (H&E staining: 20× (**A**–**P**), 40× (**B**–**Q**), and 100× (**C**–**R**) magnification).

**Figure 9 vaccines-12-00017-f009:**
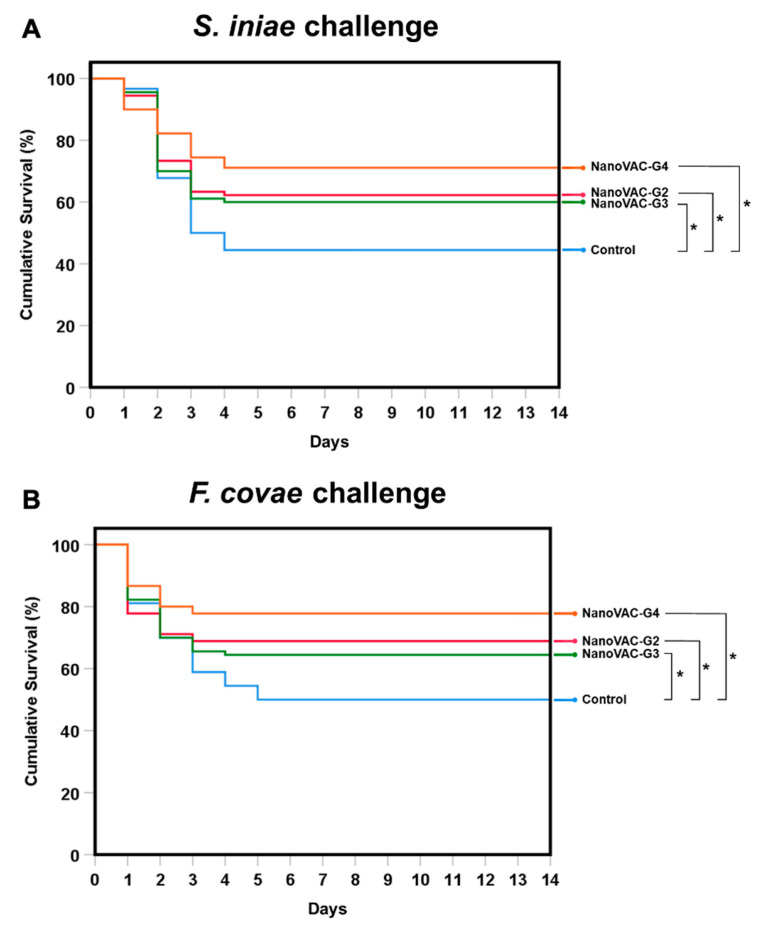
Survival analysis of Asian seabass vaccinated with bivalent nanovaccines after challenge with the single pathogens *S. iniae* (**A**) and *F. covae* (**B**). Data and survival plots were generated using the Kaplan–Meier method. The levels of statistical significance between the control and treatment groups are indicated by * (*p* < 0.05) (*n* = 30).

**Table 1 vaccines-12-00017-t001:** Primers used in this study for determining immune-related gene expression.

Primer Names	Genes	Nucleotide Sequences (5′ → 3′)	Annealing Temperature (°C)	Product Size (bp)	Accession Number
*Lc_β-actin*	β-actin	F-5′-TACCCCATTGAGCACGGTATTG-3′ R-5′-TCTGGGTCATCTTCTCCCTGTT-3′	60	160	XM_018667666.1
*Lc_IgM*	Immunoglobulin M (*IgM*) (secreted form)	F-5′-TGTCAAGGTAAACGAGGGAGC-3′ R-5′-TCCCCTGGATCCATTCGTCA-3′	60	152	ASM164080v1
*Lc_IgT*	Immunoglobulin T (*IgT*) (secreted form)	F-5′-GAGGCAACTTACAGAGGAACCATA-3′ R-5′-CTGGTCACTTCTCCATCAATTTCC-3′	60	194	ASM164080v1
*Lc_IgD*	Immunoglobulin D (*IgD*) (membrane-bound form)	F-5′-GAGTGTGAATGTTGCTGGGC-3′ R-5′-TTGGCCTGAAAGGTGACGTA-3′	60	150	ASM164080v1
*Lc_CD4*	CD4 receptor (*CD4*)	F-5′-AGTGCAATGGATTGGGGTAGATAA-3′ R-5′-GTTGCAGGCTCTGTAACTTTGATT-3′	60	156	XM_018672258
*Lc_TCRα*	T-cell receptor alpha (*TCRα*)	F-5′-GGCCGTTCGGATAGAAGGAG-3′ R-5′-AGAGCCATTGTGTTCACCGT-3′	60	153	ASM164080v1
*Lc_MHCIIα*	Major histocompatibility complex class IIα (*MHCIIα*)	F-5′-TTCCTACCTCCCTGATCTACCC-3′ R-5′-CTGAAGTCGCTGTTGGAGTAGT-3′	60	178	ASM164080v1

**Table 2 vaccines-12-00017-t002:** Characterization of *S. iniae* and *F. covae* nanovaccines.

Bacterial Culture	Formulation	Average Diameter (nm)	Zeta Potential (mV)
*Streptococcus iniae*	Sonicated antigen (bacterial cells)	203 ± 10	−36.87 ± 0.93
	Polymeric nanovaccine (solution form)	246 ± 16	45.39 ± 1.31
	Polymeric nanovaccine (dry form)	304 ± 25	47.60 ± 0.96
*Flavobacterium covae*	Sonicated antigen (bacterial cells)	324 ± 6	−21.86 ± 0.89
	Polymeric nanovaccine (solution form)	394 ± 14	38.25 ± 1.06
	Polymeric nanovaccine (dry form)	426 ± 18	36.48 ± 1.02

**Table 3 vaccines-12-00017-t003:** Growth performance of Asian seabass 30 days after vaccination with bivalent nanovaccines (DAV).

Growth Parameters	Treatments			
	Control	NanoVAC-G2	NanoVAC-G3	NanoVAC-G4
Weight gain (WG, g)	0.542 ± 0.16	0.545 ± 0.17	0.552 ± 0.12	0.544 ± 0.15
Specific growth rate (SGR, %/day)	7.18 ± 0.75	7.18 ± 0.72	7.24 ± 0.79	7.19 ± 0.74
Average daily gain (ADG, g/fish/day)	0.0181 ± 0.0016	0.0182 ± 0.0018	0.0184 ± 0.0014	0.0181 ± 0.0011
Feed conversion ratio (FCR)	1.85 ± 0.44	1.83 ± 0.37	1.81 ± 0.39	1.84 ± 0.43

## Data Availability

The data that support the findings of this study are available on request from the corresponding authors.

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
