# Peer review of "Development of Immersion and Oral Bivalent Nanovaccines for Streptococcosis and Columnaris Disease Prevention in Fry and Fingerling Asian Seabass (Lates calcarifer) Nursery Farms"

_vaccines, 2023, doi:10.3390/vaccines12010017_

Round 1
Reviewer 1 Report
Comments and Suggestions for Authors
In this manuscript " Development of Immersion and Oral Bivalent Nanovaccines for Streptococcosis and Columnaris Disease Prevention in Fry and Fingerling Asian Seabass (Lates calcarifer) Nursery Farms", the authors developed an oral bivalent nanovaccines for Streptococcosis and Columnaris disease prevention. The subject addressed in this article is worthy of investigation. Abundant experiments were conducted in this manuscript, and the result was convincing.
Several concerns:
The vaccine preparation process is complex. And the vaccine preparation conditions are too harsh, including a freeze-drying process and ultra low temperature storage conditions,
Line 150, "the final concentrations of both bacteria at 2 × 106 CFU/mL", what is the basis for this concentration? I think this concentration is too low.
The vaccination time is confused. At 30 DAH, one immersion, at 40 DAH, one immersion, at 50 DAH, one oral vaccine for 3 consecutive days. At 10 DAV immersion, day 10, day 20, day 30, which day? Authors should clearly define.
Section 3.5, RPS of Bivalent Nanovaccines need to provide.
Several figures, significant differences, letter (a, b, c) and asterisk (*) repeat.
Comments on the Quality of English Language
No
Author Response
Reviewer 1#
In this manuscript " Development of Immersion and Oral Bivalent Nanovaccines for Streptococcosis and Columnaris Disease Prevention in Fry and Fingerling Asian Seabass (Lates calcarifer) Nursery Farms", the authors developed an oral bivalent nanovaccines for Streptococcosis and Columnaris disease prevention. The subject addressed in this article is worthy of investigation. Abundant experiments were conducted in this manuscript, and the result was convincing.
Several concerns:
The vaccine preparation process is complex. And the vaccine preparation conditions are too harsh, including a freeze-drying process and ultra low temperature storage conditions.
Response: Thank you so much for this comment. We understand the intention of the reviewer well. To prevent the decline of vaccine quality, we have to strictly follow the protocol of Kitiyodom et al. (2019) [18], which detailly describes the benefit of all used techniques to meet the acceptable quality of nanovaccines.
Line 150, "the final concentrations of both bacteria at 2 × 106 CFU/mL", what is the basis for this concentration? I think this concentration is too low.
Response: Thank you so much for this suggestion. Even though the concentration of vaccine seems to be low, based on the preliminary trial, the prolonged immersion periods for 24 h are excellent enough to increase IgM levels, as indicated in section 2.3. This protocol effectively compensates for high-concentration immersion vaccination with a short immersion period.
The vaccination time is confusing. At 30 DAH, one immersion, at 40 DAH, one immersion, at 50 DAH, one oral vaccine for 3 consecutive days. At 10 DAV immersion, day 10, day 20, day 30, which day? Authors should clearly define.
Response: Thank you so much for this concern point. Oral vaccination was 3-consecutively conducted at DAHs 50, 51, and 52, as described in section 2.5. Additionally, 10 DAV immersion means day 10 after 1st immersion vaccination.
Section 3.5, RPS of Bivalent Nanovaccines needs to be provided.
Response: Thank you so much for this comment. We have added this information in “Materials and Methods” and “Results” in section 3.5.
Several figures, significant differences, letter (a, b, c) and asterisk (*) repeat.
Response: Thank you so much for this comment. We have removed the asterisks in Figure 1 and Figure 2. Additionally, the description of significant differences in these figures has already been revised.

Reviewer 2 Report
Comments and Suggestions for Authors
Dear Editor,
The manuscript entitled “Development of Immersion and Oral Bivalent Nanovaccines for Streptococcosis and Columnaris Disease Prevention in Fry and Fingerling Asian Seabass (Lates calcarifer) Nursery Farms” by Pakapon Meachasompop et al. describes the administration of bivalent nanovaccines of Streptococcus iniae and Flavobacterium covae in Asian sea bass. The nanovaccines were administered by immersion vaccination at 30 and 40 days after hatching (DAH), and the third vaccination was orally administered by feeding at 50 DAH. Following, the experimental groups immune responses were measured and discussed.
Τhe manuscripts’ objectives are quite interesting, the manuscript is well-written and could be accepted for publication after minor revisions. My detailed comments for the authors to consider are provided below:
1. The abstract should contain some information about the studied fish species, the diseases and the nanovaccines formulation (e.g. chitosan based) and not just describing the study findings. The following sentence should be omitted: Five different groups were designed: group 1 was set as the 16 control group, and groups 2-4 were vaccinated using the same protocols ‘.
2. Page 2, lines 57-68: please add some information on what immune responses are desired (e.g. Ig production) and at what developmental stage the vaccination is more efficient regarding the fish immune system maturity.
3. Page 2, lines 72-80: please add some information for the diseases, i.e. which fish species are affected, mortality rates, etc.
4. Page 3, lines 109-110: please add the microscope type and provide the staining protocol or an appropriate reference.
5. Figure 2: lysozyme and bactericidal activity have been put in the wrong order, based on the main text.
6. Page 12, lines 434 & 437: please add some arrows to the figures to point on some blue point cells and provide some more information for the immune-involved-like cells
7. I think reference 3 is irrelevant to the manuscript content. Please confirm or point the relevant information it provides.
Author Response
Reviewer 2#
The manuscript entitled “Development of Immersion and Oral Bivalent Nanovaccines for Streptococcosis and Columnaris Disease Prevention in Fry and Fingerling Asian Seabass (Lates calcarifer) Nursery Farms” by Pakapon Meachasompop et al. describes the administration of bivalent nanovaccines of Streptococcus iniae and Flavobacterium covae in Asian sea bass. The nanovaccines were administered by immersion vaccination at 30 and 40 days after hatching (DAH), and the third vaccination was orally administered by feeding at 50 DAH. Following, the experimental groups immune responses were measured and discussed.
Τhe manuscripts’ objectives are quite interesting, the manuscript is well-written and could be accepted for publication after minor revisions. My detailed comments for the authors to consider are provided below:
- The abstract should contain some information about the studied fish species, the diseases, and the nanovaccines formulation (e.g. chitosan-based) and not just describe the study findings. The following sentence should be omitted: Five different groups were designed: group 1 was set as the control group, and groups 2-4 were vaccinated using the same protocols ‘.
Response: Thank you so much for this comment. We have correctly revised this part by following the reviewer's comments.
- Page 2, lines 57-68: please add some information on what immune responses are desired (e.g. Ig production) and at what developmental stage the vaccination is more efficient regarding the fish immune system maturity.
Response: Thank you so much for this suggestion. We have correctly revised this part by following the reviewer's comments.
- Page 2, lines 72-80: please add some information for the diseases, i.e. which fish species are affected, mortality rates, etc.
Response: Thank you so much for this comment. We have correctly revised this part by following the reviewer's comments.
- Page 3, lines 109-110: please add the microscope type and provide the staining protocol or an appropriate reference.
Response: Thank you so much for this comment. We have adequately revised this part by following the reviewer's comments.
- Figure 2: lysozyme and bactericidal activity have been put in the wrong order, based on the main text.
Response: Thank you so much for this comment. We have appropriately revised this part by following the reviewer's comments.
- Page 12, lines 434 & 437: please add some arrows to the figures to point on some blue point cells and provide some more information for the immune-involved-like cells.
Response: Thank you so much for this comment. We have correctly indicated red arrows representing immune-involved-like cells.
- I think reference 3 is irrelevant to the manuscript content. Please confirm or point out the relevant information it provides.
Response: Thank you so much for this comment. We have correctly changed it with a proper reference.

Reviewer 3 Report
Comments and Suggestions for Authors
This was a very good paper and well designed study. However, I suggest the authors put in a table in the methods section detailing any differences between the three nanovaccine treatments (2-4). It was unclear in the paper how and if these differed.
Line 138 - add parenthetically "data not shown" for the preliminary study (also do for line 142).
Line 142 could be improved since I did not understand what "to scramble to" means. Assume it was top coated then mixed in well, but needs to be written in a clearer form.
Section 2.6.3 please provide the method for RNA purification.
Line 298 add again "data not presented"
Figure 3 blow up y-axis on IgD, MCH1a, TCRa and CD4 graphs to make clearer to the reader.
Need to make conclusion on line 623-626 more conditional since you did not have a non-nanovaccine control (meaning the bacterins by themselves not in nanoparticle form).
Otherwise, a very enlightening paper and well done study.
Author Response
Reviewer 3#
This was a very good paper and well-designed study. However, I suggest the authors put in a table in the methods section detailing any differences between the three nanovaccine treatments (2-4). It was unclear in the paper how and if these differed.
Response: Thank you so much for this comment. We understand and agree well with the reviewer’s suggestion. Technically, treatments 2-4 were treated with the same protocol and vaccination but located at different locations on the same nursery farm, where we taught that they must be affected by vaccination responses.
Line 138 - add parenthetically "data not shown" for the preliminary study (also do for line 142).
Response: Thank you so much for this comment. We have added this information at the reviewer’s suggestion.
Line 142 could be improved since I did not understand what "to scramble to" means. Assume it was top coated and mixed in well, but it needs to be written in a clearer form.
Response: Thank you so much for this comment. We have modified this information as the reviewer suggested.
Section 2.6.3 please provide the method for RNA purification.
Response: Thank you so much for this comment. We have modified this information as the reviewer suggested.
Line 298 add again "data not presented"
Response: Thank you so much for this suggestion. We have added this information at the reviewer’s recommendation.
Figure 3 blow up y-axis on IgD, MCH1a, TCRa and CD4 graphs to make clearer to the reader.
Response: Thank you so much for this comment. We have modified this information as the reviewer suggested.
Need to make a conclusion on lines 623-626 more conditional since you did not have a non-nanovaccine control (meaning the bacterins by themselves not in nanoparticle form).
Otherwise, a very enlightening paper and a well-done study.
Response: Thank you so much for this suggestion. In this section, we added recent information studied by our research group to support the reviewer's request.
